# Association between the Health Belief Model, Exercise, and Nutrition Behaviors during the COVID-19 Pandemic

**DOI:** 10.3390/ijerph192315516

**Published:** 2022-11-23

**Authors:** Keagan Kiely, William A. Mase, Andrew R. Hansen, Jessica Schwind

**Affiliations:** 1Water’s College of Health Professions, Department of Health Sciences and Kinesiology, Georgia Southern University, Statesboro, GA 30458, USA; 2Department of Health Policy and Community Health, Jiann-Ping Hsu College of Public Health, Georgia Southern University, Statesboro, GA 30458, USA; 3Department of Biostatistics, Epidemiology & Environmental Health Sciences, Jiann-Ping Hsu College of Public Health, Statesboro, GA 30458, USA

**Keywords:** COVID-19, pandemic, exercise, nutrition, Health Belief Model, epidemiology

## Abstract

Introduction: The COVID-19 pandemic has affected our nation’s health further than the infection it causes. Physical activity levels and dietary intake have suffered while individuals grapple with the changes in behavior to reduce viral transmission. With unique nuances regarding the access to physical activity and nutrition during the pandemic, the constructs of Health Belief Model (HBM) may present themselves differently in nutrition and exercise behaviors compared to precautions implemented to reduce viral transmission studied in previous research. The purpose of this study was to investigate the extent of exercise and nutritional behavior change during the COVID-19 pandemic and explain the reason for and extent of this change using HBM constructs (perceived susceptibility, severity, benefit of action, and barriers to action). Methods: This study used a cross-sectional design to collect 206 surveys. This survey collected information on self-reported exercise and nutrition changes during the pandemic and self-reported levels of the HBM constructs. Results: Findings showed individuals with medium or high exercise behavior change had greater odds of increased HBM score than individuals with little to no exercise behavior change (OR = 1.117, 95% CI: 1.020–1.223, SE: 0.0464, *p* = 0.0175). There was no association between nutritional behavior change and HBM score (OR = 1.011, 95% CI: 0.895–1.142, *p* = 08646). Conclusion: Individuals who reported a more drastic change in either exercise had greater odds of increased feelings of perceived susceptibility and severity related to COVID-19 and decreased perceived benefits and increased barriers to exercise. This relationship was not found regarding nutrition behavior change. These results encourage public health practitioners to understand how an individual’s perceived feelings about a threat may affect exercise and nutritional behaviors.

## 1. Introduction

COVID-19, the disease caused by SARS-CoV-2, has devastated the United States since the first known case was confirmed at the beginning of 2020. As of 19 September 2022, there were 92,296,142 confirmed cases and 1,030,010 deaths, demonstrating how the pandemic’s impact is still being felt in the social, economic, and health sectors [1,2]. There were many immediate public health initiatives and strategies implemented in the early course of the pandemic in an attempt to slow the spread of this novel disease. The health safety strategies that were promoted immediately, such as donning masks and handwashing, required relatively little additional effort to incorporate into daily living. Other strategies demanded more extensive behavior change, such as avoiding social gatherings, self-isolation, and shutting down businesses, especially those deemed ‘high risk’ by the Center for Disease Control and Prevention [3]. The effects of the virus and its associated public health impact have not been fully realized and are still being explored [2].

Gyms and fitness centers were among businesses and operations classified as moderate- to high-risk for infection [3]. This risk classification prompted many gyms and fitness centers to reorient operations to align with public health guidance, such as social distancing patrons by spreading out exercise equipment, mask mandates, and in some cases, complete closure of the facility. The combination of increased infection risk and operational changes led to subsequent changes in exercise behaviors of those previously utilizing exercise facilities [4,5,6,7,8]. The presence of disrupted exercise behaviors during the pandemic is inarguable, but the extent of this effect on exercise patterns has varied [5,6,9,10,11].

Studies examining physical activity over the past few years have consistently demonstrated a decrease in physical activity levels during the COVID-19 pandemic [4,6,9,10,12,13,14]. This change in behavior was observed in all populations when “shelter in place” orders, change in facility operations, orders to avoid social gatherings, and the overarching fear of susceptibility in public settings presented new barriers to maintaining previous exercise levels [4,6,9]. Even younger individuals, more resilient to behavior change, experienced changes in exercise patterns, with a decrease in physical activity seen on university campuses [5,15]. While most studies have reported a decrease in the total volume of exercise reflective of less frequency, duration, or intensity, some studies have further identified changes in training styles and modalities, with a decrease in resistance training and increase in aerobic activities [5,6,10]. Decreases in exercise was associated with other health implications, especially an individual’s mental health [16,17,18]. While this shift to increased aerobic training, may have been driven by the limited access to equipment required for resistance training, examination of an individual’s motivations to exercise have been shown to influence the individual’s exercise levels and modalities [10]. This decrease in motivation to exercise and other concurrent worsening health issues, such as increases in eating disorders, depression levels, pre-pandemic physical activity levels, and negative situational perspective, were all probable contributors to a change in exercise [6,10,19,20,21,22]. Amongst the individuals self-reporting changes in exercise behaviors, there was a small portion reporting increases in exercise during the pandemic, furthering the need to understand the underlying cause of the variation in exercise behavior change [10,20,22]. The consistent health guidance to participate in adequate physical activity was interrupted by the new, more relevant, health safety guidance to minimize virus transmission, placing individuals in a conundrum of balancing their physical activity level and decreasing their chances of infection.

Not all businesses could cease operation to slow the spread of this new threat. Certain workers and operations were designated as “essential” if they “conducted a range of operations and services in industries that are essential to ensure the continuity of critical functions in the United States (U.S.)” [23]. Included in the food and agriculture sector were operations that enabled the selling of human food, animal food, and beverage products at grocery, pharmacy, and convenience stores [24]. This classification and subsequent concerted effort to keep grocery stores open, while simultaneously increasing access through delivery and pick-up services, allowed for continual access to nutritional needs for most with minimal barriers outside of those presented by the health threat of the virus. Although access to food was not drastically limited such as that of exercise, research indicated altered eating habits during the pandemic [10,13,14,25,26,27]. These studies showed the changes were heterogeneous and more nuanced than the documented physical activity and exercise changes. A few studies captured the increase in alcohol, sugar-sweetened beverages, unhealthy snacks, and an overall increase in volume of food consumed [14,25,28,29]. While the dietary habits of many suffered, some individuals appeared to be unaffected or even improved their nutrition, potentially influenced by the increased time spent at home [26,27,29,30,31]. Comparable to many of the other effects of this pandemic, this phenomenon was not unique to the United States, as similar dietary behaviors were found across various populations in multiple countries [28,32,33]. The provision of grocery and food pick-up and delivery allowed for continual access to dietary needs for a greater population, an action not applicable to physical activity. Without the complete dissolution of the food supply, factors contributing to changes in nutritional behaviors during COVID-19 may unveil lessons useful for promoting healthy behaviors in the future.

A commonly used model to explain one’s proclivity to accepting health promoting behaviors is the Health Belief Model (HBM). Contained in the HBM are four distinct constructs that reflect one’s feeling on the perceived health threat and perceived outcome of “taking action”: susceptibility, severity, benefits, and barriers [34,35]. These constructs are independent of each other and their ability to provide explanation of partaking in health behaviors makes the HBM ideal for investigating various health behaviors during the pandemic. The HBM is widely used in cross-sectional studies looking at predicting various health-promoting or disease-preventing behaviors [35]. HBM has been used previously to examine behavior during the COVID-19 pandemic, indicating a relationship between belief of perceived threat and having better adherence to health promoting behaviors [36,37,38]. The inclusion of self-efficacy in the HBM framework has shown to increase an individual’s likelihood to “take action”, despite low perceived risk [39]. With unique nuances regarding the access to physical activity and nutrition during the pandemic, the constructs of HBM may present themselves differently in nutrition and exercise behaviors compared to precautions implemented to reduce viral transmission studied in previous research.

While the literature indicated an evident change in nutrition and exercise behaviors, there was a paucity of research quantifying the extent of the change in exercise and nutrition behavior during the pandemic. Further, the elucidation on how various HBM constructs may have affected someone’s readiness or likelihood to change nutritional and exercise behaviors could benefit future pursuits to improve these outcomes. Therefore, the purpose of this study was to investigate the extent of exercise and nutritional behavior change during the COVID-19 pandemic and explain the reason for and extent of this change using HBM constructs.

## 2. Methods

### Study Design and Sampling Methods

This study used a cross-sectional design with snowball sampling. Researchers developed a survey using Google Forms to acquire all information. The survey was piloted by a small group representative of the study population but unaffiliated with the research project. All material was reviewed and approved by the Institutional Review Board (#H22104). The survey was then introduced to university students and faculty members with the directions to send the survey to other members in the community. The survey was administered during the timeframe of October 2021 through January 2022.

Among the 206 responses collected, the mean age was 29 years old with a range of 18–81 years of age. Most participants were within healthy BMI range (mean = 25.83 kg/m^2^, SD = 5.66). Participants were 66% female (n = 136) 33.5% male (n = 69) and 0.5% chose not to respond (n = 1). The majority (75.24%) of individuals identified as White (n = 155), 17% identified as Black (n = 35), and approximately 8% identified as another race/ethnicity (n = 16). A total of 83 reported little to no exercise behavior change, while 62 and 61 reported medium and high change, respectively. The individuals who self-reported medium to high changes were statistically significantly more likely to be stressed, have lower number of healthy habits, lower self-efficacy, and HBM score. Although not statistically significant, individuals with a greater exercise change were more likely to have a higher BMI and had a greater proportion of females and black and other races. The distribution of independent variables across outcome are reflected in Table 1.

## 3. Dependent Variables

### 3.1. Exercise Behavior Change

All changes in exercise behavior were assessed using a Likert scale ranging from 0 (no change) to 5 (drastic change). The questions were worded in a manner that implied the change was a negative change. Utilizing the same exercise components examined by Steele et al. [6], face validity was used to create four questions measuring the change in exercise frequency, intensity, duration, and motivation. The following questions were used to evaluate the change in exercise frequency, intensity, duration, and motivation:Frequency: Since the onset of the pandemic, the frequency of exercise has decreased.Intensity: Since the onset of the pandemic the intensities of my workouts have decreased.Duration: Since the onset of the pandemic, the durations of my workouts have decreased.Motivation: Since the onset of the pandemic, I have been less motivated to be physically active.

The (response categories) to the questions were worded in a way that generated a higher score to represent a larger extent of negative exercise behavior change. The mean response for all questions was calculated, and the final score was a continuous variable between 0 and 5. The change in exercise behavior was categorized into the following categories: 0–1 = little to no change, >1 and ≤3 = moderate change, >3 = high change.

### 3.2. Nutritional Behavior Change

All changes of nutritional behavior were assessed using a Likert scale ranging from 0 (no change) to 5 (drastic change). Eight questions were generated to gather information on changes in alcohol, sugar, fast-food, fruits/vegetables, and processed food consumption, as well as the individuals’ attitude about the “healthfulness” of their diet during COVID-19. The eight questions were structured based on the lifestyle habit questions produced by Kumari et al. [40] that were shown to have strong internal validity and consistency.

Fruit and vegetable intake: Since the onset of the pandemic, I have decreased the amount of vegetables and fruits I eat.Healthfulness: Since the onset of the pandemic, my nutritional choices have become less healthy.Fast-food consumption: The amount of fast-food I eat has gone up.Alcohol consumption: The amount of alcohol I consume has gone up.Diet composition awareness: The awareness of my food consumption has decreased.Energy intake: I consume more daily calories than I did before the pandemic.Processed food consumption: The amount of processed foods has increased.Sugar-sweetened beverages: The amount of sugar sweetened beverages I consume has increased.

The questions were worded in a way that generated a higher score to represent a larger extent of negative nutrition behavior change. The mean response for all questions was calculated, and the final score was a continuous variable between 0 and 5. Based on the data and context, researchers dichotomized nutrition change into 0–1 = little to no change and >1 = moderate to significant change.

## 4. Independent Variables

### 4.1. Demographics

Participants were asked to provide information on their age in years, height in inches, weight in pounds, race, and their sex. Height and weight information were used to generate the individual’s body mass index (BMI). Age and BMI were treated as continuous variables in the analysis, while race was categorized as White, Black, and Other.

### 4.2. Health Belief Model

Four questions investigated the following aspects of the HBM: susceptibility, severity, benefit, and barriers [35]. The questions created to examine perceived severity and susceptibility were generated using the modified WHO Cosmo protocol questionnaire [41]. Questions were based on a 0–5 Likert scale and were worded in a way that higher scores would reflect greater health beliefs related to COVID-19. The questions were as follows:Perceived susceptibility: I believe I am susceptible to contracting COVID-19 (If you have been vaccinated, this question refers to before vaccination). (0 = I am not susceptible, 5 = I am highly susceptible).Perceived severity: I believe if I become infected with COVID-19 I will have a severe infection (If you have been vaccinated, this question refers to before vaccination). (0 = I will not have a reaction, 5 = I will have a very severe reaction).Perceived benefits: I believe that exercise during the pandemic has large benefits in decreasing my risk of COVID-19. (0 = It will not benefit me, 5 = Exercise will benefit me against the risk of COVID-19.Perceived Barriers: I believe that COVID-19 pandemic has presented many barriers to exercise. (0 = There have been no barriers, 5 = There have been many barriers).

The responses were recorded as a continuous variable and individual questions were summed to derive a HBM score (range: 0–20). The higher the score corresponded to higher of the individual’s perceived severity, susceptibility, barriers, and a lesser benefit of exercise. Research showed high levels of these constructs correspond to greater participation in health-promoting behaviors [36]. In the context of this study the health-promoting behaviors associated with high levels of these constructs were behaviors that decreased likelihood of transmission, such as high-risk facilities or avoiding situations in which social distancing or donning a mask was difficult. These behaviors may be health-promoting in one context but may also negatively affect one’s exercise and nutritional behaviors.

### 4.3. Exercise Self-Efficacy

Self-efficacy is defined as one’s confidence in their abilities to successfully perform a particular behavior and is associated with one’s ability to engage in behavior change [42]. Individuals who focus on self-efficacy within the framework of Social Cognitive Theory (SCT) believe both knowledge and confidence are important factors in behavior change [43]. The two questions were generated to evaluate the knowledge and confidence needed to effectively adapt exercise behaviors to the new COVID-19 environment.

Confidence: I have not felt confident in my ability to adapt my exercise routine to the changing environment that COVID-19 has presented.Knowledge: I feel I have a good amount of knowledge about exercise which helps me participate in exercise.

The first question was asked to assess the individual’s confidence to adapt their exercise behaviors to the COVID-19 environment. The second question was used to assess the individual’s knowledge of exercise. The scores of the second question were reverse coded so the scores of the questions could be added together with a high score representing lower exercise self-efficacy. Both questions used a 0–5 Likert scale in which higher numbers on the scale were associated with lower confidence and knowledge, after the reverse coding. The responses of both questions were used to generate a score of exercise self-efficacy.

### 4.4. Pre-Pandemic Physical Activity

One question recorded the participants’ pre-pandemic physical activity. The participant was asked to choose one of the following choices that best described their physical activity levels prior to March 2020.

Below recommendation: Less than 150 min of moderate to vigorous activity a week or 1–2 workouts.Meets recommendation: 150–300 min of moderate to vigorous activity a week or 3–5 workouts.Exceeds recommendation: Over 300 min of moderate to vigorous activity a week or over 5 workouts.

Physical activity was defined as “any bodily movement produced by skeletal muscle that requires energy” and is not restricted to only intentional exercise within a fitness setting [41]. This variable was dichotomized as either did not meet recommended or met/exceeded recommended levels of physical activity. Met and exceed expectations were combined due to the low observation of those who exceeded the recommendation, allowing for better statistical efficiency.

### 4.5. Healthy Habits

Individuals were asked to choose healthy habits that were a part of their lifestyle. The four healthy habits consisted of:4.I do not smoke.5.I do not consume more than 2 alcoholic beverages a day (12 ounces of regular beer, 5 ounces of wine, 1.5 ounces of distilled spirits).6.I get 150 min of moderate to vigorous physical activity a week.7.I consume at least 1 serving of fruit or vegetables a day (1 serving is equal to a cup or the size of an adult closed fist).

The summation of total number of healthy habits chosen were used to create a discrete variable reflecting the total number of healthy habits but the specific healthy habits were undistinguishable.

### 4.6. Stress

To identify the level of self-reported stress the individual experienced during the pandemic a single question using a Likert-scale was used to gauge stress from 0 (no stress) to 5 (dramatically increased stress).

The COVID-19 pandemic has added stress to my life.

This variable was dichotomized with responses of 0–2 representing “no to low stress” and 3–5 representing “mild to high stress”. This dichotomization allowed for greater statistical efficiency due to the small sample sizes observed across some levels.

## 5. Statistical Analysis

Descriptive statistics stratified across outcome variables were compared using the chi-square test for categorical variables and analysis of variance (ANOVA) for continuous variables across the three groups. Ordinal logistic regression was used to examine the association between the HBM score (summation of all four constructs), and exercise behavior change across the three categories. Logistic regression was used to examine the association between the HBM score and the nutritional behavior change outcome. Another logistic regression was run to evaluate which of the four individual HBM construct had the greatest association with the outcome variable. The Hosmer and Lemeshow method for variables selection was used for all models [44]. All independent variables were included in the initial model and variables with a *p*-value > 0.25 were removed from the model. Backwards selection was then used by eliminating any variable that was not associated to the outcome with a *p*-value < 0.05. If the removal of a variable caused another variable’s coefficients to change more than 10%, then the latter variable was retained as the model was sensitive to its inclusion. All variables were tested for interaction and collinearity. All model assumptions were met, and all statistical tests were performed using SAS 9.4 software with statistical significance set at *p* < 0.05.

## 6. Results

The final logistic regression model chosen using Hosmer and Lemeshow’s intentional variable selection demonstrated a statistically significant relationship between exercise behavior change and HBM scores after adjusting for self-reported stress, exercise self-efficacy, and nutrition behavior change (X^2^ (4) 70.7611, *p* < 0.0001). While controlling for other variables, it was observed that individuals with medium or high exercise behavior change had greater odds of increased HBM score than individuals with little to no exercise behavior change (OR = 1.117, 95% CI: 1.020–1.223, *p* = 0.0175). There were also increased odds of exercise behavior change with increasing stress (OR: 1.245, 95% CI: 1.003–1.544, *p* = 0.0467), increasing self-efficacy score (representing lower self-efficacy levels) (OR: 1.509, 95% CI 1.255–1.814, *p* < 0.0001), and nutrition behavior change (OR: 2.371, 95% CI 1.800–3.123, *p* < 0.0001). All other variables were not included due to lack statistical significance and little to no impact on the coefficients of other variables after removal of these variables. The model is shown in Table 2.

The binary regression model was statistically significant after adjusting for self-reported stress, age, healthy habits, pre-pandemic physical activity level, and exercise behavior change (X^2^ (6) 49.8328, *p* < 0.0001). After controlling for other variables, there was no association between nutritional behavior change and HBM score (OR = 1.011, 95% CI: 0.895–1.142, *p* = 08646). There was a statistically significant positive relationship in which those who reported greater changes in exercise had greater odds of higher levels of stress (OR: 1.395, 95% CI: 1.065–1.826, *p* = 0.0154) and exercise behavior change (OR: 1.168, 95% CI: 1.093–1.248, *p* < 0.0001). A negative relationship was found between age (OR: 0.929, 95% CI: 0.900–0.959, *p* < 0.0001), health habits (OR: 0.674, 95% CI: 0.451–1.008, *p* = 0.0547), and pre-pandemic physical activity levels (OR: 0.221, 95% CI: 0.095–0.515, *p* = 0.0005). These findings are reflected in Table 3.

An ordinal logistic regression was used to distinguish which component of the HBM accounted for the largest proportion of variability in exercise behavior change. All four components of the HBM (susceptibility, severity, benefits, and barriers) were included in a single model. Due to the specific interest in these four variables, no other variables were included in the model. The overall model was statistically significant (X^2^ (4) 43.5539, *p* < 0.0001) but within the model, barriers was the only variable found to be statistically significant, indicating those with a greater change in negative exercise behaviors had a greater odds of reporting higher levels of perceived barriers to exercise (OR: 1.977, 95% CI: 1.609–2.428, *p* < 0.0001). The output is reflected in Table 4.

## 7. Discussion

The purpose of this study was to investigate the extent of negative changes in exercise and nutrition behavior during the COVID-19 pandemic and further describe the extent of this change using HBM constructs. We observed that those who reported a more drastic change in either exercise intensity, duration, frequency, and/or motivation had greater odds of increased feelings of perceived susceptibility and severity related to COVID-19 and decreased perceived benefits and increased barriers to exercise. The racial distribution across outcome groups varied with a greater proportion of individuals that had a high degree of change in exercise behaviors change begin Black or a race other than White (*p* = 0.137). A greater proportion of individuals who reported medium to high exercise change were females (*p* = 0.265). Individuals with greater negative change in exercise behaviors were also more likely to have higher self-reported stress and BMI (*p* < 0.0001 and *p* < 0.08, respectively). The odds of 0.929 decreased with increasing age, and only remained statistically significant in reference to nutritional behavior change. This finding may be explained by younger individuals (18–30 years old) who tend to report lower perceived threat from COVID-19 [45]. Males were also less likely to report behavior change compared to females, which aligns with gender differences seen in adherence to health precautions throughout the pandemic [46,47,48,49]. Individuals with a BMI classified as overweight/obese were more likely to report a change in exercise behavior. This finding may reflect the abundant precautions taken by overweight/obese individuals due to the increased severity of infection in overweight/obese individuals [50,51,52].

Within our study, 60% of respondents had moderate to high levels of exercise behavior change and 40% had little to no change. This proportion varied among studies which may be due to populations and cut-off differences [4,6,9,13,14,22]. As stated earlier, this decrease in physical activity could have deeper effects on an individual’s physical and mental health [16,17,18]. We observed that with each increase in HBM score the odds of a greater extent of exercise behavior change increased by 11.7%. This association was influenced by an individual’s self-reported stress, exercise self-efficacy, and nutrition behavior, so these variables were included in the final model. Exercise frequency was the most impacted due to COVID-19, with 74% of participants stating they had some degree of change. Steele et al. [6] found that individuals who exercise 4–5 days per week had the greatest decrease in exercise frequency compared to pre-pandemic periods. Similarly, Brand et al. [20] found that pandemic exercise frequency was dependent on pre-pandemic levels, while our study showed no relationship between the two. This absence of an association in our study may be explained, in theory, by individuals working out >5 days a week have a habitual routine they strive to maintain and others with <3 days a week may continue to seldomly work out when possible. Exercise duration, compared to intensity and frequency, had the greatest number of responses stating individuals had no change during the pandemic. This finding suggests that once someone was able to adapt their exercise routine to the changing situation, they continued with the same intensity and duration, leading to only a small proportion of respondents reporting changes in these areas. In some studies, nonactive individuals increased their exercise [10,20,22]. Since we did not allow for individuals to mention if they had increased exercise, we were unable to observe this phenomenon.

Perceived barriers had the greatest influence on exercise behavior change. However, perceived susceptibility, severity, and benefits did not have any statistical significance association with exercise behavior change. Our finding that barriers provided the greatest influences on exercise behavior change does not come with much surprise. Steele et al. [6] reported just how much of an impact the closure of fitness and recreational facilities had on exercise behaviors during the “lockdown” period highlighted. Further, Gildner et al. [19] found that over 90% of respondents were currently “sheltered in place”, which the authors believed was the underlying cause of decreased exercise behavior. Perceived barriers was shown in previous cross-sectional and prospective studies to be the most powerful hinderance in the acceptance of other healthy behaviors [35,45,53], and susceptibility to the health threat was a poor predictor of health behaviors [35,45,53]. Trifiletti et al. [54] further supported this observation in the current pandemic with finding that the combination of susceptibility and severity, often referred to as perceived threat, did not have a strong association with hand washing and social distancing adherence. Moreover, studies prior to the pandemic elucidated the weak association of perceived severity and behavior change, possibly further weakened in our study by the low perceived severity and susceptibility of COVID-19 in younger generations [15,35,37,52]. Though, it is important to note high levels of self-efficacy can promote healthy behaviors, despite the perception of low risk or susceptibility [11,39]. The weak association between perceived threat of a highly transmissible and deadly disease is puzzling and may have valuable public health implications in efforts to produce behavior change.

This study observed that individuals with the greatest decrease in exercise had greater odds of lower self-reported self-efficacy. These findings reiterate previous research that demonstrated increased self-efficacy had a significant positive effect on a breadth of health outcomes in various populations [52]. Von ah and colleagues [45] reported the adaptability of self-efficacy to be a mediating or moderating variable in the association of barriers and health behaviors, highlighting the importance of its inclusion in depictions of health behaviors. Naturally, if one feels competent in one’s ability, they will have a higher probability of positive behavior change when faced with an obstacle or barrier, because they are confident they can perform the desired behavior. This observation may explain why individuals in our sample who maintained and did not reduce their exercise, had a greater odds of exercise self-efficacy. Unlike early adapted precautions of the pandemic, self-efficacy cannot be enhanced instantly and requires various influences (e.g., vicarious experiences and learned knowledge) to build over time. This finding supports the need for methods of creating self-efficacy of health-promoting behaviors throughout an individual’s lifetime.

The current study observed approximately 50% of respondents had dietary changes reflecting a change in alcohol, fruit, vegetable, and sugar-sweetened beverage consumption and an overall increase in the volume of food intake. The current body of literature showed a similar proportion of nutritional change [10,13,14,25,26,27]. The most affected aspects of diet seemed to be healthfulness of diet, fast-food consumption, and overall calories consumed, with the latter two aspects likely driving the overall decreased healthfulness. Our results demonstrated that 70% of respondents reported their dietary changes as unhealthy, which was a distinct difference to the 19% of individuals who reported unhealthy changes in Park and colleagues [25]. Phillipou et al. [13] found a majority of people had relatively unchanged eating habits during the pandemic, but 35% of individuals experienced increased binge eating, a troubling trend our study did not investigate. Along with the aforementioned nutritional changes, Chen et al. [14] showed an increase in eating out and alcohol consumption in young adults, an age group reflective of a large proportion of our sample. A similar proportion of participants reported changes in exercise behaviors and may explain the moderate correlation between exercise and nutrition scores (r = 0.538). Interestingly, HBM did not have a significant association with nutrition behavior. This difference may be because the same barriers that were imposed on exercise behaviors were not imposed on nutrition behaviors.

The current study observed that the total number of reported healthy habits acted as a protective factor for nutrition change, further supporting the protective relationship between pre-pandemic habits and resistance to drastic nutritional changes during the pandemic. This finding adds to the literature as other studies reviewed did not examine pre-pandemic healthy habits. We also observed a protective effect of pre-pandemic physical activity levels among respondents who had little to no change in nutrition behavior having greater odds of meeting physical activity recommendation pre-pandemic. Combined with the results of self-efficacy, it seems vital that individuals strive to create and maintain healthy lifestyle behaviors, for the immediate benefits as well as the benefit of increased resilience when exposed to situations or environments that make healthy physical activity and nutrition behaviors more challenging to achieve.

## 8. Strengths

The strength of this study lies in the examination of exercise behavior change in regard to an individual’s perceived susceptibility and severity of COVID-19, benefits and barriers to exercise. This study effectively showed how changes in these factors were associated with a negative change in exercise behaviors. A unique characteristic of this study was the association of healthy habits and their protective factors on exercise behaviors. Previous research has shown that healthy habits (i.e., vegetable consumption, adequate exercise levels, etc.) were shown to have significant health benefits regardless of an individual’s BMI [55,56]. Although small, the sample contained a wide range of ages and BMIs, and gender and race had adequate distribution across outcome categories.

## 9. Limitations

There are study limitations to consider when interpreting the results. The proportion of various demographic variables, specifically the average age of the sample, restrict the generalizability of the results as this may influence the perceived threat of the virus [37] With all research, the results should always be interpreted and generalized in the context of the sample in the study. Specific to our study, the sample contained a relatively high proportion of young adults. This study also did not allow for the capturing of individuals who may have improved nutrition and exercise behaviors during the pandemic. The questions in the survey only allowed for two options, no change, or some extent of negative change. Although this was done with the intention of examining the possible factors that contributed to a decrease in healthy lifestyle behaviors, it may have introduced bias if individuals did not feel like their behavior change was adequately represented in the response choices. Lastly, the self-reporting of “health” may be distorted due to the absence of a well-defined, objective parameter. For example, a large portion of people in this study reported their diet became less healthy. We cannot deduce from this finding what was unhealthy about their nutritional decisions and what, if any, health implications this unhealthfulness would present. Without the objective metric, the reporting of less healthy nutritional habits was subject to the individual’s perceptions, feelings, and ideas of healthy nutrition.

## 10. Public Health Implications

From the initial onset of the COVID-19 pandemic, public health authorities operationalized the precautionary principle with respect to population health recommendations. Traditional communicable disease risk reduction strategies of social distancing, hand washing, masking, covering coughing, and syndromic surveillance were employed, but were often not enough to reduce the epidemic curve. During the initial year of the pandemic, case fatality rates were high, and hospital and healthcare workforce capacity was insufficient to keep up with the demand for healthcare services. With this insufficient infrastructure, greater emphasis was stressed on “at-risk” individuals isolating and reducing risk of infection. The same comorbidities putting these individuals at risk were highly prevalent in individuals who lacked proper physical activity levels and dietary habits. This study illustrates the need to promote dynamic, evidence-based health promotion policy implementation, educating the public as to risks and benefits to pro-health behavior change and decision-making. Health behavior change, as illustrated by these analyses, needs to be implemented through a careful consideration of both costs and benefits with a future-focused lens for the downstream effects resulting from health policy implementation.

## 11. Conclusions

Our research sought to examine if differing levels of self-reported HMB constructs had a negative impact on exercise and nutrition behaviors during the COVID-19 pandemic. Our study demonstrates the effects that self-reported HBM constructs have on exercise and nutrition behaviors during the COVID-19 pandemic. Higher perceived susceptibility, severity, and barriers combined with a decreased belief in the benefit of exercise was associated with greater decreases in exercise behavior but not dietary intake. Higher levels of exercise self-efficacy were shown to be protective against negative changes in exercise behaviors. As the pandemic continues and other health threats appear, understanding an individual’s perceived threats and barriers to engaging in physical activity is vital to identify individuals that may be vulnerable to a decreased physical activity level. This study, and the pool of literature in which it adds to, demonstrates the need for a concerted effort of public health, fitness, and wellness practitioners to form a concerted effort to increase access and adherence physical activity during times of increased barriers and worries of a health threat. This needs to be addressed as it has a far-reaching impact on many different aspects of an individual’s and communities’ health and well-being.

## Figures and Tables

**Table 1 ijerph-19-15516-t001:** Demographic Characteristics of Participants Across Exercise Behavior Change Classifications.

Characteristic	Little to No (n = 83)	Medium (n = 62)	High ^a^ (n = 61)	*p*-Value
Sex				
Male	33 (40.24%)	18 (29.03%)	18 (29.51%)	
Female	49 (59.76%)	44 (70.97%)	43 (70.49%)	0.265 ^c^
Race				
White	68 (81.93%)	48 (77.42%)	39 (63.93%)	
Black	9 (10.84%)	10 (16.13%)	16 (26.23%)	0.137 ^d^
Others	6 (7.23%)	4 (6.45%)	6 (9.84%)	
Age	31.48 (14.50)	27.76 (14.40)	28.38 (12.64)	0.225 ^b^
BMI (kg/m^2^)	24.9 (4.69)	25.7 (5.33)	27.11 (6.91)	0.080
Self-reported Stress	2.95 (1.54)	3.59 (1.27)	4.03 (1.24)	<0.0001
Health Belief Model Score	10.33 (3.05)	11.74 (3.19)	12.46 (3.44)	0.0004
Exercise Self-efficacy	5.24 (1.59)	5.55 (1.24)	6.82 (1.64)	<0.0001
Healthy Habits	3.23 (0.941)	2.98 (1.016)	2.64 (0.86)	0.0013

^a^. Exercise Behavior Change Classifications: Little to No = <1, Medium >1 and <3, High >3. ^b^. An analysis of variance (ANOVA) was used for all continuous variables. ^c^. A chi-square test was used for categorical variable. ^d^. A Fisher’s exact test was used when cells contained less than 5 observations. Significance was set at alpha = 0.05.

**Table 2 ijerph-19-15516-t002:** Health Belief Model and Exercise Behavior Change: Ordinal Logistic Regression Model.

	Adjusted Odds Ratio	95% CI	*p*-Value
Health Belief Model Score	1.117	1.020–1.223	0.0175 *
Self-reported Stress	1.245	1.003–1.544	0.0467
Exercise Self-efficacy	1.509	1.255–1.814	<0.0001
Nutrition Score	2.371	1.800–3.123	<0.0001

* Significance was set at alpha = 0.05.

**Table 3 ijerph-19-15516-t003:** Health Belief Model and Nutrition Behavior Change: Binary Logistic Regression Model.

	Adjusted Odds Ratio	95% CI	*p*-Value
Health Belief Model Score	1.011	0.089–1.142	0.865 *
Self-reported Stress	1.395	1.065–1.826	0.015
Age	0.929	0.900–0.959	<0.0001
Healthy Habits	0.674	0.451–1.008	0.055
Pre-pandemic Physical Activity	0.221	0.095–0.515	0.0005
Exercise Score	1.168	1.093–1.248	<0.0001

* Significance was set at alpha = 0.05.

**Table 4 ijerph-19-15516-t004:** Health Belief Model Components and Exercise Behavior Change: Ordinal Logistic Regression Model.

	Adjusted Odds Ratio	95% CI	*p*-Value
Susceptibility	1.012	0.821–1.249	0.908 *
Severity	1.000	0.792–1.262	0.999
Benefits	0.947	0.789–1.136	0.557
Barriers	1.977	1.609–2.428	<0.0001

* Significance was set at alpha = 0.05.

## Data Availability

The data presented in this study are available on request from the corresponding author.

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
