# Peer review of "Association between the Health Belief Model, Exercise, and Nutrition Behaviors during the COVID-19 Pandemic"

_ijerph, 2022, doi:10.3390/ijerph192315516_

Round 1

Reviewer 1 Report

The article presents sufficient quality and rigor. However, it would be advisable for the authors to strengthen the introductory section by adding more studies this would make the manuscript more solid and meaningful. The authors here could link from the literature correlation between the decrease in physical activities and its impact on Psychological and mental safety. Moreover, this could be linked to the impact of maintaining physical activity on the educational capability of students.

It is recommended that the conclusions section could be added.

Author Response

Dear Reviewer 1,

We would like to thank you for taking the time to read our study and add crucial suggestions to strengthen it and allow us to contribute to this prestigious journal. We appreciate the acknowledgment of quality and rigor our paper has. Much consideration went into the introduction, and we believe your suggestions further strengthen it. We have added a few lines in the introduction and discussion mentioning the implications decreased physical activity has had on mental health during the COVID-19 pandemic. Again, thank you for your suggestions.

We have added a conclusion section to the end of our manuscript. This section summarizes the entirety of the paper in simple terms to help readers leave with a full understanding of the material discussed. Your suggestion is appreciated and has made an impactful contribution to the paper.

Reviewer 2 Report

The present study is devoted to the analysis of the level of changes in behavioral responses during exercise and rational food consumption during the spread of the coronavirus pandemic. The authors attempted to explain the reason for this phenomenon using the application of HBM constructs. The study used a special self-diagnosis of participants in the experiment on the analysis of physical activity and nutrition during a pandemic. The results of the monitoring showed that study participants with medium and high arousal in the emotional state during motor activity had higher rates for the risk of an increase in the HBM score than participants with lower rates. However, the study did not record any relationship in the indicator of changes in rational and proper nutrition during physical exertion.

In a peer-reviewed study, the authors proved that healthy habits in a person's life act as an important protective factor in conjunction with proper nutrition between the period before the pandemic and nutrition during the spread of coronavirus infection. This study is of particular practical importance in that there are few similar studies in terms of analyzing the healthy habits of people of different ages.

Despite all the positive aspects of the study, the authors did not fully disclose the significance of the study results for the practice of proper nutrition and specific types of physical activity. The study would benefit from a detailed explanation of the criteria and indicators for changes in mental behavior in study participants during exercise.

Reviewer 3 Report

If the sample is not homogenous, it is even more important to interpret the findings in detail. (and that is missing in your paper).

Line 11: At the beginning of the abstract, in the one or two sentences at the top, it should be explicated why the reader should be interested, including any historical references.

Line 13: Cannot put an abbreviation for the first time without indicating its meaning (HBM)

Line 17: Methods: Disclose which analytical methods are being used to collect data; describe participants and instrument closer.

Line 23: Conclusion: Include the implications of the presented research and what it adds to the selected field as a whole.

Line 136: Study Design and Sampling Methods.

Please identify the characteristics of a sample in a presented survey (gender, age, faculty, etc.)

Describe and justify your choice of sampling strategy. Regardless lines 296-306: should go to the section “Study Design and Sampling Methods.”

Line 307: Table 1. Sex, Race should be at the top of the table, followed by age, etc.,

                 All the tables should include the level of significance (below the table).

Line 357-358, 362, 365, 404, 419, 426, 438: Please specify your findings (the differences between age and gender, race) closely.

Line 467: You should also include the strength of the presented investigation.

Line 468: The last section should also include a restatement of the research problem and a

                 summary of findings.

Line 485: It seems like not all the references from the text are included in the Reference 

                Section.

It is advisable to consider a more detailed explanation of the interplay between gender, age, and education in at least five HMB components.
